# Parental Educational Intervention to Facilitate Informed Consent for Pediatric Procedural Sedation in the Emergency Department: A Parallel-Group Randomized Controlled Trial

**DOI:** 10.3390/healthcare10122353

**Published:** 2022-11-23

**Authors:** Yen-Ko Lin, Yung-Sung Yeh, Chao-Wen Chen, Wei-Che Lee, Chia-Ju Lin, Liang-Chi Kuo, Leiyu Shi

**Affiliations:** 1Division of Trauma and Surgical Critical Care, Department of Surgery, Kaohsiung Medical University Hospital, Kaohsiung Medical University, Kaohsiung 80708, Taiwan; 2Department of Medical Humanities and Education, College of Medicine, Kaohsiung Medical University, Kaohsiung 80708, Taiwan; 3Department of Emergency Medicine, College of Medicine, Kaohsiung Medical University, Kaohsiung 80708, Taiwan; 4Center for Medical Education and Humanizing Health Professional Education, Kaohsiung Medical University, Kaohsiung 80708, Taiwan; 5College of Nursing, Kaohsiung Medical University, Kaohsiung 80708, Taiwan; 6Department of Health Policy and Management, Bloomberg School of Public Health, Johns Hopkins University, Baltimore, MD 21205, USA

**Keywords:** informed consent, pediatric procedural sedation, knowledge, satisfaction, emergency department

## Abstract

Obtaining valid parental informed consent for pediatric procedures in the emergency department (ED) is challenging. We compared a video-assisted informed consent intervention with conventional discussion to inform parents about pediatric procedural sedation in the ED. We conducted a prospective randomized controlled trial using a convenience sample including the parents of children in the ED in whom procedural sedation for facial laceration was recommended. The video group watched an informational video. Conventional group participants received information from physicians during conventional discussion. The primary outcome was knowledge improvement of the video intervention compared with conventional discussion. The secondary outcome was parental satisfaction. Video and conventional groups comprised 32 and 30 participants, respectively. Mean knowledge scores of parents after intervention [±standard deviation] were higher in the video group (91.67 ± 12.70) than in the conventional group (73.33 ± 19.86). Knowledge score differences were significantly bigger in the video group (coefficient: 18.931, 95% confidence interval: 11.146–26.716). Video group participants reported greater satisfaction than conventional group participants. Parents’ comprehension of and satisfaction with the informed consent process for pediatric procedural sedation may be improved with the use of an educational video. Standardized approaches should be developed by healthcare institutions to better educate parents, facilitate treatment decisions, and boost satisfaction in the ED.

## 1. Introduction

For ethical and legal reasons, it is crucial that physicians provide patients with information about procedures, risks, benefits, and treatment alternatives during the informed consent process [1,2,3]. Obtaining informed consent is a process, not merely a matter of documentation [4,5,6,7,8,9]. Patients must understand the relevant information to provide their consent for medical procedures. A full understanding of such information helps patients to make informed individual choices [10,11,12,13].

Obtaining legitimate informed consent in the emergency department (ED) is a difficult process in that it usually takes longer to communicate and discuss relevant information with patients or their families [14]. Providing parents or surrogates with adequate knowledge prior to emergency procedures in children can be particularly challenging. The lack of time, physical pain, and emotional distress usually create demanding situations for patients and families when it comes to understanding the important information needed to make treatment decisions and provide consent in emergency settings [15,16,17,18,19]. Therefore, physicians must make a particular effort to communicate information to patients and families and assist them in making well-informed decisions, even under such difficult conditions.

Procedural sedation involves the use of sedative, analgesic, and dissociative drugs to produce analgesia, sedation, and motor control during uncomfortable or painful diagnostic and therapeutic procedures [20]. It is an ethical imperative to provide children with relief from pain and anxiety associated with diagnostic and therapeutic procedures; from the parents’ perspective, such relief is an indicator of the quality of care [20,21]. Healthcare institutions should develop policies and procedures to safely deliver procedural sedation [22]. Although the safety profile of procedural sedation has improved, patients are still exposed to considerable risks from potentially severe adverse events. Reported rates of adverse events range from 0.6% to 17% and include airway complications (e.g., partial airway obstruction, respiratory depression, apnea, laryngospasm), emesis, and recovery agitation (defined as any combination of agitation, crying, hallucinations, or nightmares) [20,21,23,24,25,26,27].

During the traditional informed consent process in the ED, patients and their family members may struggle to understand how the treatment procedure or surgery will progress as well as how to make treatment decisions; this is because a large amount of information is provided to them during this process [15,17]. Therefore, using an educational video to support the process of informed consent might be a practical approach. Numerous studies have revealed that using video to educate patients improves patient knowledge concerning the risks and complications of procedures and promotes patient satisfaction [28,29,30,31,32,33,34]. There may be both an ethical and practical need for a video-assisted intervention to facilitate informed consent for pediatric procedural sedation. In this study, we explored the use of an educational intervention to improve the informed consent process. Some studies have shown that interventions to educate patients and their families help them to retain more information [28,29,33]. However, to the best of our knowledge, the use of video-assisted informed consent before pediatric procedural sedation in the ED has never been investigated.

The aim of the present study was to develop an educational video intervention tool to assist the informed consent process and to determine whether using an educational video could improve routine discussion with parents about the risks, benefits, and alternatives of procedural sedation for their children in the ED.

## 2. Materials and Methods

### 2.1. Intervention Tool

The conceptual framework and study design, developed by Lin et al. [35] to evaluate video-assisted informed consent for adults after trauma in the ED setting, were modified and applied to this study. In this study, we developed a video and knowledge assessment questionnaire. At the start of the study, At the study commencement, a panel of experts referred to the published literature and best practice guidelines for pediatric procedural sedation [20,21,23,24,25,26,27] to develop the knowledge assessment questionnaire and video content. Eight experts with different expertise were invited, including two emergency physicians, one anesthesiologist, one pediatrician, two nursing practitioners, one patient’s parent who had experienced the pediatric procedural sedation, and one member from the ethics committee. Every expert was selected based on the recommendations of two specialists in our university hospital. The expert panel met regularly to reach a consensus regarding these materials. The video was created and produced by the Center for Development of Multimedia Digital Material of Kaohsiung Medical University. The final script contained information about the sedation procedure, risks, benefits, and alternatives. Two-dimensional graphics software and role-playing by actors were used to develop the educational video. Written subtitles and captions were added, and audio narration was used to support the video content. The video included several sections: Introduction, Purpose, and Benefits of the Procedure, Preparation and Monitoring, Procedure, Complications and Risks, and Questions and Answers. Treatment alternatives were included in the Questions and Answers section.

A portable computer preloaded with the information program was used, and the volume was appropriately adjusted so that participants could hear the content. A research associate was present during the participant’s navigation through the program to ensure completion and help as needed. Watching the entire video took approximately 6 min, after which participants could ask a physician questions about the procedure.

A questionnaire assessing knowledge regarding the purpose, benefits, and risks of procedural sedation was compiled based on previous questionnaires [20,21,23,24,25,26,27]. The questionnaire was also used to collect participant data, including age, sex, and education level. The questionnaire was tested on 5 parents whose children received procedural sedation in our ED. Some sentences and wording had been modified. The expert panel revised the questionnaire after a pilot test to produce a final version (see Appendix B).

### 2.2. Study Design

During the second stage of the study, we conducted a prospective randomized controlled trial using a superiority study design in a tertiary university hospital ED. Study participants included the parents of children recommended for procedural sedation for a facial laceration. Ketamine (4–5 mg/kg, intramuscular) was provided for pediatric procedural sedation. Participants randomized to the video group received the video-assisted informed consent intervention, which explained the procedure and its risks, benefits, and alternatives. The conventional group participated in routine discussion with a physician, who provided information about the procedure. All participants completed the knowledge questionnaire (Appendix B) before and after the educational sessions. The questionnaire comprised three questions (Appendix B), rated on a 5-point Likert scale, used to evaluate respondents’ satisfaction with the informed consent procedure. Before the study began, the institutional review board of the hospital evaluated and approved the study protocol. Participants provided their written informed consent before enrollment.

### 2.3. Participant Selection

All parents of pediatric patients older than 1 year and younger than 7 years of age in whom procedural sedation for facial laceration had been recommended were eligible for enrollment if a trained research associate was available. Exclusion criteria included parental refusal to participate, parental inability to understand the study process, patient clinical instability, or parents’ inability to speak and read Mandarin. If an eligible participant was missed, the reason, such as the unavailability of the research associate, was recorded in a study logbook. 

The following power assumptions were made: (a) higher education levels increase the mean questionnaire score (post-knowledge scores will be higher; mean difference 16% for the conventional group and 32% for the video group); (b) scores are normally distributed; (c) standard deviation is 16 for both groups; (d) level of significance is 0.05 (*p* < 0.05); (e) *t*-test is two-tailed; and (f) 20% dropout rate is used to analyze the data. Given these assumptions, we determined that a sample size of 29 in each group was needed to achieve an effect size of more than 0.8 with 90% power and a significance of 0.05. 

### 2.4. Data Collection and Processing

A prescript approach was used to enroll eligible participants in the study, and written informed consent was obtained from all participants. Simple randomization was applied. A computer-based random number generator was used to produce odd or even values, and accordingly, participants were randomly assigned to either the conventional or video group. The allocation procedure was concealed. After randomization, a research associate collected basic characteristics (age, sex, and self-reported education level). Additional information was collected from the hospital’s electronic medical records system and patient documents, such as arrival time to the ED and emergency physician. 

All participants completed the multiple-choice knowledge assessment before the educational sessions. Conventional group participants received information about the procedure from their physician in a conventional discussion. Following the discussion, conventional participants completed the knowledge assessment again and responded to questions assessing their satisfaction with the informed consent process. Participants in the video group watched the informational video on a laptop computer at the patient’s bedside. Participants could pause and/or replay the video. All video participants received the question-and-answer session after watching the video. If participants had additional questions about the procedure, they could speak with a physician after the educational session. The aim of this question-and-answer session was to provide a similar opportunity for questions as that provided to the conventional group during the informed consent process. The same knowledge assessment and satisfaction survey were completed by participants in both groups after the question-and-answer session.

In our ED, informed consent for the procedure was obtained by emergency attending physicians. Physicians who obtained informed consent were not aware of the questions in the knowledge assessment. The research associate was blinded to the allocations when assessing the outcomes.

The primary outcome was the participants’ understanding of pediatric procedural sedation as evaluated using the knowledge assessment. Questions were multiple-choice and equally weighted (Appendix B). The maximum total score was 6 points, and scores were converted to a percentage (total possible knowledge score range: from 0% to 100%). Secondary outcomes were participant satisfaction with the informed consent process (rated on a 5-point ordinal Likert scale) and the frequency of consent refusal.

### 2.5. Data Analysis

Data were recorded using participant numbers, with no individual identifying information to maintain participant anonymity. Collected data included participant demographics, ED arrival time, and the treating physician. Demographics of the conventional and video groups were analyzed using descriptive statistics. Continuous variables were analyzed using mean and standard deviation, and categorical variables were analyzed using proportions. Binary, ordinal, and categorical variables were calculated using Fisher’s exact test. The Student *t-*test was used to compare mean pre- and post-education scores on the knowledge questionnaire between groups. The paired *t-*test was used for within-group comparisons.

We calculated the knowledge score difference by subtracting pre-education scores from post-education scores. If independent factors were associated with knowledge score difference and satisfaction in univariate analysis or if these were clinically significant, they were included in the multivariable regression models. We used two regression models with predefined covariates for knowledge score differences, including a multiple linear regression model and a multiple ordinal logistic regression model. The knowledge score differences were categorized and applied to the analysis of the multiple ordinal logistic regression model. A multivariable ordinal logistic regression model with preset factors for parental satisfaction was used. Likelihood ratio tests were used for the multivariable models. A threshold of *p* < 0.05 was used to indicate statistical significance, and 95% confidence intervals were computed. Stata version 14.0 (StataCorp LP, College Station, TX, USA) was used to analyze relevant data.

## 3. Results

During the 1-year study period, 118 children received procedural sedation (Figure 1). Forty-five participants were not enrolled because no research associates were available. Four parents who were unable to understand the study process and seven who refused to participate were excluded (Figure 1). Thus, the data for 62 participants were analyzed, 32 were allocated to the video group, and 30 were allocated to the conventional group. There were no significant differences in basic characteristics between the conventional and video groups (Appendix A).

### 3.1. Knowledge Scores

Table 1 summarizes the main results for knowledge scores. There was no statistically significant difference in baseline knowledge scores between the video and conventional groups. Knowledge scores were higher after the educational sessions (mean knowledge scores 91.67 for the video group versus 73.33 for the conventional group) than at baseline (mean knowledge scores 54.69 for the video group versus 52.78 for the conventional group) in both groups, but participants in the video group had higher post-education knowledge scores than those in the conventional group (mean knowledge scores 91.67 versus 73.33, respectively). After the educational sessions, knowledge scores in the video group (mean knowledge score difference: 36.98) increased more than those of the conventional group (mean knowledge score difference 20.00) (Table 1). We categorized knowledge score differences as ≤20, 21–49, and ≥50; there were significant between-group differences in scores in these categories (Table 2). 

In the subgroup analysis of baseline knowledge scores, there was no statistically significant difference between the conventional and video groups (Appendix A). Post-education knowledge scores for the conventional and video subgroups were compared, and scores were significantly greater for some video subgroups (Appendix A), including female sex, arrival time other than 08:00–16:00 h, and physicians B and C. Regardless of age (<34 or ≥34 years) and education level (<college or ≥college), participants in the video group had significantly higher post-education knowledge scores.

Table 2 shows the results of the subgroup analysis for knowledge score differences. For the subgroups age < 34 years, female sex, education level college and above, arrival time other than 08:00–16:00 h, and physicians B and C, knowledge score differences for the video group were significantly higher than those for the conventional group. Because the above subgroups showed greater knowledge score differences, participants in those subgroups may have been more impacted by video education. 

Table 3 shows the analysis results of knowledge score differences using different regression models, including a simple regression model, multiple linear regression model, and multiple ordinal logistic regression model. We investigated the adjusted impact of video-based education using a multivariate linear regression model while controlling for predetermined factors. The findings showed that, following video education, there was a considerable increase in knowledge score differences; on average, these gaps rose by 18.931 points. The baseline knowledge score had a significant impact on the knowledge score differences (coefficient: 0.766); a lower baseline knowledge score was correlated with greater knowledge score differences. In the multiple ordinal logistic regression model, the adjusted odds ratio revealed that the intervention increased knowledge score differences (the adjusted odds ratio for knowledge score difference in the video group was 21.243). 

### 3.2. Parental Satisfaction

Table 4 shows the results regarding participant satisfaction. We found statistically significant differences between the video and conventional groups for the statements, “I understand the procedural information provided by health care providers”, “The information provided by health care providers helped me in making a choice regarding the procedure”, and “The informed consent process for the procedure is satisfactory”. No participants refused to give their consent for the procedure.

Table 5 shows the multivariable ordinal logistic regression model results for parental satisfaction, controlling for the predefined covariates of age, sex, education level, arrival time, physician, and baseline knowledge score. The adjusted odds ratio for the video group indicated that the video improved perceptions of satisfaction. The adjusted odds ratios for “I understand the procedural information provided by health care providers”, “The information provided by health care providers helped me in making a decision regarding the procedure”, and “The informed consent process of the procedure is satisfactory” were 4.838 (95% confidence interval 1.606–14.575), 3.871 (95% confidence interval 1.312–11.422), and 6.544 (95% confidence interval 2.088–20.507), respectively. The adjusted odds ratio for the education level indicated that the education level had an impact on perceptions of satisfaction. The adjusted odds ratios of education level for “I understand the procedural information provided by health care providers” and “The information provided by health care providers helped me in making a decision regarding the procedure” were 4.278 (95% confidence interval 1.072–17.066) and 5.178 (95% confidence interval 1.305–20.548) respectively.

## 4. Discussion

We found that participants who received a video-assisted informed consent intervention had greater knowledge of pediatric procedural sedation. Referring to the multimedia principle, a theory of multimedia learning, Mayer proposed that people learn at a deeper level through pictures and words than through words only [36]. Compared with the conventional group who had conventional discussions, participants in the video intervention group had greater levels of satisfaction with the informed consent procedure. As far as we know, this was the first study using an educational video to enhance the informed consent process for pediatric procedural sedation in the ED.

The two core principles of patient autonomy and patient well-being are promoted in the informed consent process [15,17], and physicians must respect and support these principles. Informed consent encompasses several essential components, including competence, disclosure, and voluntariness [37,38]. A competent patient must receive adequate information from healthcare providers, understand the risks and benefits, and make decisions voluntarily based on the patient’s own values. In addition to ethical aspects, the legal prerequisites of informed consent require physicians to provide information about medical procedures, benefits, risks, complications, and available alternatives [15,39]. 

It is difficult and time-consuming to obtain legitimate informed consent in the ED. Owing to time limits, strong emotions, and stress brought on by pain or acute symptoms, it is frequently hard for patients and their family members to comprehend and take in sufficient information to be able to give their informed consent for emergency procedures [15,16,17,19]. Parents of children in the ED are generally considered legal surrogates and decision-makers for pediatric patients. It is presumed that parents will make decisions according to the best interests of their child, maximizing the benefits and minimizing possible risks and suffering [40]. Therefore, healthcare providers must devote effort to effectively communicating the required information to patients, thereby enabling them to make decisions under difficult circumstances.

Research on the decision-making framework for informed consent has identified many factors that affect patients’ understanding of the process. Patient factors (e.g., age, education level, previous experience), physician factors (e.g., years in clinical practice, personal communication skills, usage of educational aids), disease context (e.g., disease type, disease severity), and environmental factors (e.g., ordinary or emergency settings, time of the visit) may affect information exchange, patient deliberation, and voluntarism regarding treatment decisions and consent [41,42]. Our findings indicate that, after controlling for several factors, the use of the educational video and participant baseline knowledge influenced parents’ knowledge and understanding. Future research is necessary to investigate whether and how these factors affect the informed consent process for pediatric patients and their parents in the ED.

A previous study reported that parental educational level and sex affect risk counseling recall; children’s mothers showed better risk recall than fathers [43]. However, another study revealed that demographics were not related to parental recall rates during informed consent for emergency surgery for their children [44]. In our study, maternal parents and parents with an education equal to or above college level in the video group showed a significantly greater knowledge score difference. Although the difference was not statistically significant in the multivariable analysis, parents’ sex and education level seemed to have a greater impact on parental understanding after the video intervention. Moreover, there was a significant difference in baseline knowledge scores between college- and non-college-educated parents. Mean baseline knowledge scores [± standard deviation] were higher in the college group (62.61 ± 14.38) than in the non-college group (40.67 ± 14.50). The vocabulary of the knowledge assessment and test-taking skills might also have influenced the results of the test. Future research is required to verify these findings.

The entire consent process is of great importance. A good consent process may improve patient satisfaction and foster a good patient–physician relationship. Several strategies have been proposed to improve the consent process in the emergency setting [16,45]. One study reported that using informational aids may improve parental satisfaction with surgical informed consent for children [46]. Another study revealed that visual aids may improve communication about surgery between pediatric surgeons and surrogates [47]. In shared decision-making, the physician can act in partnership with parents, providing information about the disease diagnosis, treatment choices, and predictive prognosis, as well as possible risks and complications [48]. Physicians should discuss parents’ perspectives and preferences, help them to discuss their values, and help them to make the best decisions for their children. Demonstration materials, booklets and pamphlets, video explanations, computer interactive programs [3,49,50,51,52,53,54,55,56,57], and “repeating back” or feedback testing [58,59,60] have all been used to increase patient and parent comprehension. 

We found that parental knowledge and comprehension were significantly predicted by baseline knowledge score and usage of an educational video. This echoed the previous study [35]. Some previous findings have indicated that the use of video to educate adult patients and facilitate informed consent in the ED is effective [35,49,61,62]. However, the effectiveness of preoperative education for pediatric patients and parents in the ED has not attracted much research interest. One study reported that a standardized presentation viewable on a portable computer was an effective way to improve knowledge and facilitate the informed consent process among the parents of children undergoing emergency surgery [63]. Traditionally, information about medical treatments or procedures is presented in written and/or verbal formats. However, research suggests that this standard approach often yields poor participant understanding of the information provided [64]. Possible reasons for poor understanding and memory retention include participants’ lack of ability to read and understand material written above the suggested reading level, rushed and incomplete disclosure from information providers, use of unfamiliar medical terminology, and variability in the clarity and amount of information. In one systematic evaluation of informed consent among trauma patients, it was shown that written and visual information improved risk retention and understanding more than verbal information alone [45]. Compared with patients who received written or verbal information, those who were provided with video information reported greater levels of satisfaction [44]. The present results show that the video-assisted informed consent intervention resulted in improved understanding compared with the conventional informed consent discussion.

Recent advancements in tablet and portable computer technologies have given rise to the excellent potential for improving preoperative parental education regarding pediatric emergency surgery [16]. The newest tablets and portable computers feature larger screens, more memory, and high image quality, making it easier to view video and instructional content. Consequently, using innovative portable computing technology might result in more efficient and timely preoperative education in the ED. In this study, we used a laptop computer with a preloaded video to educate parents and facilitate informed consent at the bedside in the ED. Our results were promising, but additional studies are needed to further explore the present findings.

We did not measure the time spent in the informed consent process for pediatric procedural sedation in the two groups. Time spent in the informed consent process may affect participant comprehension and satisfaction [43,47,59]. However, we believe that participants in both groups had sufficient time to acquire satisfactory levels of knowledge and had similar opportunities to clarify their questions with physicians so as to make medical decisions during the informed consent process. Though the consent process for each individual physician might vary, we believe it might be unlikely that time affected our results.

Furthermore, the importance of adequate education and training for healthcare providers to deliver structured and comprehensive patient information should not be underestimated [65,66]. No specific training for obtaining consent was provided for our healthcare providers, and the consent process for each individual physician might be different. Effective communication of complicated information helps patients to understand the relevant information and make decisions. A good informed consent process can also promote good patient–physician relationships and build trust.

Our study has several limitations. First, this study focused on one intervention for a particular population at one institution; thus, the findings may not be generalizable to other situations or populations. Second, although the video-assisted informed consent intervention improved participant understanding, it remains unclear how the intervention influenced the decision to provide consent for the procedure. No participants refused consent after being informed through either the conventional process or the video-based process. However, the choice of study design and population may make it more likely that participants were already aware of the need for procedural sedation and had made an initial decision to accept it prior to study recruitment. Further studies are needed to examine the effect of video-assisted informed consent intervention on decision-making. Moreover, there are no reliable and valid measures of participants’ understanding of the risks, benefits, and alternatives of pediatric procedural sedation. In this study, the video and knowledge assessment developed by a panel of experts demonstrated face validity and included information that we believed participants should know before providing consent for this particular procedure. Knowledge assessment focused on the purpose, risks, and benefits of pediatric procedural sedation, and participants showed substantial improvement in their understanding of these aspects. However, knowledge about alternative treatments was not measured. It would have been useful to assess this and to determine whether the intervention improved participants’ understanding of the alternatives. Moreover, the test question type and literacy level might have influenced the results of the knowledge assessment. However, it remains unclear as to whether the use of percentages for possible complications is easily understood and appreciated by parents in the knowledge assessment, although we believe one of the strengths of the questionnaire would be the secondary outcome of satisfaction we measured, which was more generalizable. Furthermore, seven eligible parents refused to participate. They expressed that they were too anxious to participate in the study. We did not evaluate the effect of video education on parents’ anxiety. A previous study reported that a multimedia intervention significantly reduced anxiety and increased knowledge among parents [67]. Further studies are needed to confirm these findings in the ED. More than forty percent of potential participants were missed due to the unavailability of the research associate. Whether night-time visits or other factors might have an influence on our results requires further research.

## 5. Conclusions

The use of an educational video can improve the process of informed consent for pediatric procedural sedation in the ED. Intervention using an education video may improve participant comprehension of the procedure and satisfaction with the informed consent process. To confirm these initial findings, additional studies are needed among patients receiving different types of procedures or surgeries. To better inform parents in the ED, healthcare institutions should develop organized and standardized methods for providing informed consent. Educational videos may be beneficial to improve communication among healthcare providers, patients, and families and facilitate treatment decisions.

## Figures and Tables

**Figure 1 healthcare-10-02353-f001:**
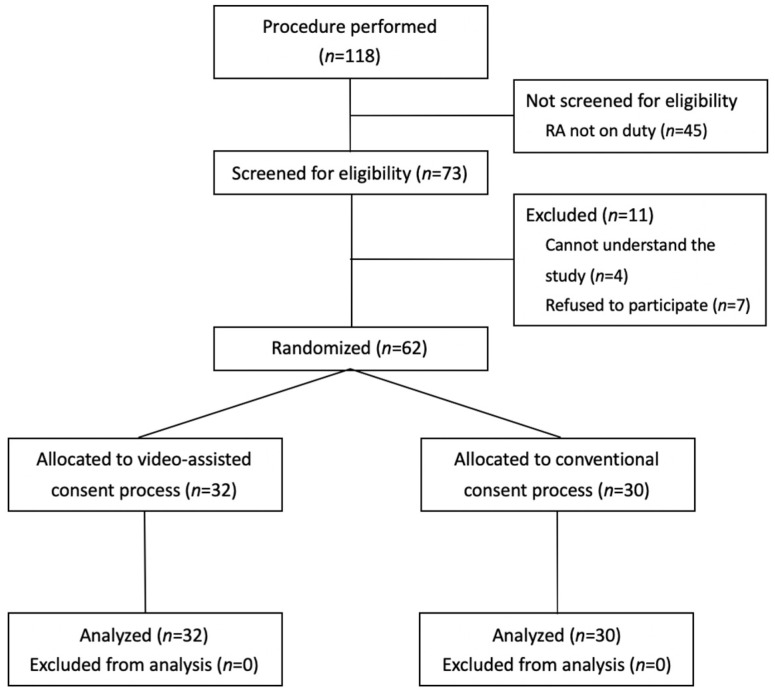
Randomized controlled trial flowchart. RA, Research associate.

**Table 1 healthcare-10-02353-t001:** Comparison of knowledge scores between conventional and video groups.

Knowledge Score	Conventional Group (*n* = 30)		Video Group (*n* = 32)		*p*-Value ^3^
	Mean	Standard Deviation	Mean	Standard Deviation	
Baseline	52.78	19.61	54.69	16.52	0.679
Post-education	73.33	19.86	91.67	12.70	<0.001
Difference ^2^	20.00	14.11	36.98	20.62	<0.001 ^1^

^1^ Unequal variance test. ^2^ Difference = (post-education score)−(baseline score). ^3^ Video versus conventional.

**Table 2 healthcare-10-02353-t002:** Subgroup analysis for the knowledge score differences.

Variable	Conventional Group			Video Group			*p*-Value	*p*-Value for Experimental Group and Variable Interaction
	*n*	Mean	Standard Deviation	*n*	Mean	Standard Deviation		
Age (years)								
<34	16	18.75	11.98	17	41.18	18.74	<0.001 ^1^	0.203
≥34	14	21.43	16.57	15	32.22	22.24	0.149 ^1^	
Sex								
Female	23	18.84	13.58	24	40.28	18.98	<0.001 ^1^	0.086
Male	7	23.81	16.26	8	27.08	23.46	0.757 ^1^	
Education								
<College	13	25.64	16.12	12	40.28	25.09	0.102 ^1^	0.611
≥College	17	15.69	10.98	20	35.00	17.85	<0.001 ^1^	
ED arrival time								
08:00–16:00 h	12	20.83	10.36	9	31.48	22.74	0.219 ^1^	0.355
Other	18	19.45	16.42	23	39.13	19.85	0.001 ^1^	
Physician								
Physician A	4	12.50	8.34	5	33.33	20.41	0.087 ^1^	0.464
Physician B	5	23.33	9.13	4	37.50	8.33	0.047 ^1^	
Physician C	4	12.50	15.96	8	43.75	21.71	0.022 ^1^	
Physician D	7	21.43	18.54	6	30.56	19.48	0.409 ^1^	
Physician E	5	26.67	19.00	6	30.56	24.53	0.774 ^1^	
Physician F	5	20.00	7.45	3	50.00	28.87	0.210 ^1^	

^1^ Unequal variance test. ED, emergency department.

**Table 3 healthcare-10-02353-t003:** Knowledge score difference evaluated using regression models.

	Simple Linear Regression Model		Multiple Linear Regression Model		Multiple Ordinal Logistic Regression Model	
	Coefficient	95% CI	Coefficient	95% CI	Odds Ratio	95% CI
Video group	16.979 ***	7.941–26.016	18.931 ***	11.146–26.716	21.243 ***	4.944–91.269
Age	−0.568	−1.492–0.356	0.104	−0.617–0.825	0.998	0.892–1.116
Sex (reference group, female)	−4.231	−15.902–7.439	−6.497	−15.470–2.477	0.556	0.143–2.156
Education (reference group < college)	−6.540	−16.633–3.553	9.286	−0.529–19.102	3.229	0.730–14.289
ED arrival time (reference group other)	5.091	−5.433–15.616	−1.132	−9.334–7.070	0.739	0.215–2.533
Physician (reference group, physician C)	−5.666	−18.287–6.955	4.511	−5.718–14.740	1.085	0.241–4.886
Baseline knowledge score	−0.539 ***	−0.784––0.294	−0.766 ***	−1.048––0.483	0.890 ***	0.844–0.938
Constant			23.093	−18.411–64.598		
			R^2^ = 0.515		Likelihood ratio test for model: *χ*2 = 44.36; *p* < 0.001	

*** *p* < 0.001. Sample size of regression model = 62. CI, confidence interval; ED, emergency department.

**Table 4 healthcare-10-02353-t004:** Comparison of satisfaction between conventional and video groups.

Outcome	Conventional Group Number (%)	Video Group Number (%)	*p*-Value
I understand the procedural information provided by healthcare providers			0.016 ^1^*
Strongly agree	11 (36.7)	22 (68.8)	
Agree	11 (36.7)	9 (28.1)	
Fair	7 (23.3)	1 (3.1)	
Disagree	1 (3.3)	0 (0.0)	
Strongly disagree	0 (0.0)	0 (0.0)	
The information provided by healthcare providers helped me in making a choice regarding the procedure			0.012 ^1^*
Strongly agree	11 (36.7)	19 (59.4)	
Agree	10 (33.3)	12 (37.5)	
Fair	9 (30.0)	1 (3.1)	
Disagree	0 (0.0)	0 (0.0)	
Strongly disagree	1 (1.4)	0 (0.0)	
The informed consent process for the procedure is satisfactory			0.006 ^1^**
Strongly agree	9 (30.0)	22 (68.8)	
Agree	14 (46.7)	9 (28.1)	
Fair	6 (20.0)	1 (3.1)	
Disagree	1 (3.3)	0 (0.0)	
Strongly disagree	0 (0.0)	0 (0.0)	

* *p* < 0.05; ** *p* < 0.01. ^1^ Fisher’s exact test.

**Table 5 healthcare-10-02353-t005:** Multiple ordinal logistic regression model for participant satisfaction.

	I Understand the Procedural Information Provided by Health Care Providers		The Information Provided by Health Care Providers Helped Me in Making a Choice Regarding the Procedure		The Informed Consent Process for the Procedure Is Satisfactory	
	Odds Ratio	95% CI	Odds Ratio	95% CI	Odds Ratio	95% CI
Video group (reference group, conventional group)	4.838 **	1.606–14.575	3.871 *	1.312–11.422	6.544 ***	2.088–20.507
Age (reference group < 34 years)	0.680	0.236–1.955	1.018	0.361–2.870	0.622	0.214–1.809
Sex (reference group, female)	1.134	0.324–3.970	1.413	0.397–5.026	0.906	0.258–3.184
Education	4.278 *	1.072–17.066	5.178 *	1.305–20.548	3.405	0.822–14.097
(reference group < college)						
Arrival time (reference group, other)	0.675	0.221–2.064	0.504	0.167–1.526	0.520	0.165–1.644
Physician (reference group, physician C)	0.917	0.211–3.976	0.503	0.106–2.394	0.495	0.102–2.402
Baseline knowledge score (reference group < 60)	0.450	0.109–1.863	0.318	0.077–1.309	0.378	0.088–1.630
Likelihood ratio test for model	*χ*2 = 15.10; *p* = 0.035		*χ*2 = 16.36; *p =* 0.022		*χ*2 = 14.83; *p* = 0.009	

* *p* < 0.05; ** *p* < 0.01; *** *p* < 0.001. Sample size of regression model = 62. CI, confidence interval.

## Data Availability

The datasets obtained and/or analyzed during the current study are available from the corresponding author upon reasonable request.

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
