# Peer review of "Parental Educational Intervention to Facilitate Informed Consent for Pediatric Procedural Sedation in the Emergency Department: A Parallel-Group Randomized Controlled Trial"

_healthcare, 2022, doi:10.3390/healthcare10122353_

Round 1

Reviewer 1 Report

I appreciate the opportunity to review the manuscript “Parenteral educational intervention to facilitate informed consent for pediatric procedural sedation in the emergency department: a parallel group randomized controlled trial”.  This study evaluates the interesting intervention of using a brief educational video to assist in obtaining informed consent for procedural sedation for facial laceration repair, in children 1-7 years of age in the emergency department.  This intervention is of clinical interest due to the potential for ensuring clear, accurate, standardized, thorough information is provided to parents at the time of obtaining informed consent. 

The paper is generally well-written , however the biggest limitation is the questionable validity of the primary outcome, the knowledge scores.  Since the control patients were receiving information from providers without familiarity with the knowledge assessment tool, it is unlikely that they would have provided the specific information included in the tool, while presumably this was explicitly covered in the interventional video.  Whether these specific percentages of incidence of complications in the knowledge test are truly essential parts of informed consent is uncertain, and thus the authors may have simply demonstrated that the parents remembered the video content.   In some cases, such as the question about fasting time without reference to what the last liquid/food ingested was, it is actually impossible for even an expert to answer the question without the video reference (i.e. the answer could be 2 hours if clear liquid, or longer if solid food was ingested).  Regardless, the increased parental satisfaction with the video intervention is important.

Specific questions/recommendations by section are listed below:

1.      Introduction: line 52, recommend changing “logical” to “well-informed”

2.      Introduction: line 67, “During the traditional informed consent process patients and their family members frequently struggle . .”.  Do you have a reference that supports this statement?  If yes, would cite after this sentence, and that would make it a much stronger rationale for the study.  If no, would change “frequently” to “may”.

3.      Methods: line 87: recommend providing slightly more information about the Lin article that provided the conceptual framework and study design, such as “developed by Lin et al to evaluate video-assisted informed consent for adults after trauma in the ED setting, were modified. . ” or similar

4.      Methods: line 90:  what best practice guidelines were used to develop the knowledge assessment and video content?

5.      Appendix A:  As mentioned, the knowledge assessment tool is the biggest limitation of the study.  It contains several examples of medical jargon (arrhythmia, gastroenteritis, aspiration), question types not recommended in the test writing community (negative question #1, all of the above question #5), a question that is impossible to answer without additional information (#4), and percentages that are too close to each other in #2 and #3.   The limitation of this tool needs to be emphasized even more in the limitations in the discussion, and the secondary outcome of satisfaction needs to be emphasized a bit more as this may be more generalizable.  For limitations, would mention among other things that it is unclear whether use of percentages for possible complications is easily understood and appreciated by families.

6.      Methods line 123-4:  I’m assuming you actually mean that the parents provided their written consent?

7.      Methods line 127: why was this age range chosen?

8.      Why was age <34 vs greater chosen to evaluate the parents?

9.      Table S2:  It looks like there was a significant difference in overall preintervention knowledge scores between college vs. non-college educated parents, not particularly surprisingly.  I think this speaks not just to general knowledge, but to the vocabulary of the test and test taking skills.   I would recommend assessing and reporting this (ie mean pretest knowledge score in college vs. non college mean, SD, p value) to make it clear.  The Flesch-Kincaid grade level of the test questions in appendix A (you can run this through Microsoft Word) is 11.9.   Meanwhile, the grade level of the questions in S4 (the satisfaction test) was 5.5.   Again I think this should be more clear in the discussion.

10.   Results:  Table 2 categories seem somewhat arbitrary, would consider removing this table.

11.   Results:  I would recommend bringing Table S4 into the main manuscript since this outcome of satisfaction is quite important and perhaps more generalizable than the knowledge test results.

12.   Table 4:  What is the binary outcome that the ordinal logistic regression model is modelling?  Would consider taking this out or explaining further in the methods and/or legend.

13.   Discussion:  Recommend adding to limitations above concerns about the knowledge assessment.  Especially the reading level, and that you may have just “taught to the test” with the video, and it is unclear whether the percentages of complications tested in the knowledge assessment are really essential parts of informed consent

Reviewer 2 Report

This manuscript describes a prospective randomized controlled trial of a video intervention to support the consent process for procedural sedation vs physician consent only. Knowledge retention and satisfaction of the parents were assessed, and showed a benefit to including video interventions. This is a study with a very small sample number and performed only in a single institution, but the authors comment on this fact in their discussion. Additionally, the control intervention is not very standardized. The introduction is informative. The materials and methods require some additional detail. The discussion can be shortened and be written more to the point, but raises many important limitations. Tables could become more informative with some minor changes. Overall, a well-designed study, that might form the base for some larger scale/multi-institutional studies. Please see specific comments below.

Specific comments:

Line 24-25: would re-word this sentence since efficacy has not been tested (unclear if it takes less time, less resources, etc.). The primary outcome variable was knowledge retention/improvement assessed by pre-and post- questionnaires

26-27: Mean knowledge score “of parents after intervention” were higher…

60: would remove; irrelevant for this manuscript.

66: reference missing

70: reference missing

78: … to retain more information. – reference/s missing

81-84: “improve routine discussion” . The study overall sounded as if patients in the video intervention arm had the option to have a discussion with a physician, but that it wasn’t additional in every case to what is routinely discussed. Would try to be more specific.

89: what is the composition of the “expert panel”? How were experts selected?

92: reference media company

108: why was revision of the questionnaire necessary, what were the results of the pilot? Any differences to presented study?

118: Did the physicians receive any standardized training? State training level (attending physician) here. Did satisfaction vary with different physicians? how many physicians were involved?

120: include reference to appendix with knowledge questionnaire here

121: include parent satisfaction questionnaire as an appendix and reference here. Re-word sentence. It sounds as if the knowledge questionnaire only included three 3 questions about satisfaction.

131: include potential reasons for missing here or just state that the only reason participants were missed was RA unavailability.

160: in the results section it would be interesting to state how many participants actually used the opportunity to talk to a physician afterwards.

163-164: Was a written consent form used for consents for both the video intervention and the physician discussion? Did the consent form contain the same information as the video intervention?

196: what were the reasons for the 7 refusals. Include in discussion.

199: reference relevant table

203: include values from results table in parentheses

209: what was the rationale behind this categorization? Was this done post-hoc or pre-defined before the study was initiated?

251-256: explain rationale behind adding "provided by health care provider". This appears misleading to those in the video group who might or might not have talked to a physician after the video instruction.

How many talked to a physician after the video intervention?

271: comment on contribution of education level to coefficients

352: missing reference

374: appears to overstate the evidence since it appears that there was no quality control for the physician consent process

376-384: unclear how this paragraph relates to the current study. Would consider removing.

389: I would recommend commenting on the fact that there was no specific training to health care providers obtaining consent.

390: include in limitations that only ½ of potential participants were included due to RA availability. Were all the night-time visits missed (there might be a difference in retention when parents are tired), etc.

Table S1: include p-values for demographics table

Table S2: indicate significant values with “*” or similar

Table 3: indicate significant values with “*” or similar

Table 4: were none of the results  of p values in the ranges of <0.05 to <0.01?
